# Physiactisome: A New Nanovesicle Drug Containing Heat Shock Protein 60 for Treating Muscle Wasting and Cachexia

**DOI:** 10.3390/cells11091406

**Published:** 2022-04-21

**Authors:** Valentina Di Felice, Rosario Barone, Eleonora Trovato, Daniela D’Amico, Filippo Macaluso, Claudia Campanella, Antonella Marino Gammazza, Vera Muccilli, Vincenzo Cunsolo, Patrizia Cancemi, Gabriele Multhoff, Dario Coletti, Sergio Adamo, Felicia Farina, Francesco Cappello

**Affiliations:** 1Department of Biomedicine, Neuroscience and Advanced Diagnostics, University of Palermo, 90127 Palermo, Italy; rosario.barone@unipa.it (R.B.); eleo.trovato@gmail.com (E.T.); damicoda90@gmail.com (D.D.); claudia.campanella@unipa.it (C.C.); antonella.marinogammazza@unipa.it (A.M.G.); felicia.farina@unipa.it (F.F.); francesco.cappello@unipa.it (F.C.); 2Department of Neuroscience, Cell Biology, and Anatomy, University of Texas Medical Branch, Galveston, TX 77573, USA; 3SMART Engineering Solutions & Technologies Research Center, eCampus University, 22160 Novedrate, Italy; fil.macaluso@gmail.com; 4Euro-Mediterranean Institutes of Science and Technology, 90139 Palermo, Italy; 5Department of Chemical Sciences, University of Catania, 95129 Catania, Italy; v.muccilli@unict.it (V.M.); vcunsolo@unict.it (V.C.); 6Department of Biological Chemical and Pharmaceutical Sciences and Technologies, University of Palermo, 90127 Palermo, Italy; patrizia.cancemi@unipa.it; 7Department of Radiation Oncology, School of Medicine, Central Institute for Translational Cancer Research, Technical University of Munich, TranslaTUM, 80333 Munich, Germany; gabriele.multhoff@tum.de; 8DAHFMO Unit of Histology and Medical Embryology, Sapienza University of Rome, 00185 Rome, Italy; dario.coletti@uniroma1.it (D.C.); sergio.adamo@uniroma1.it (S.A.); 9Biological Adaptation and Ageing, CNRS UMR 8256, Inserm ERL U1164, Institut de Biologie Paris-Seine (IBPS), Sorbonne Université, 75005 Paris, France

**Keywords:** cachexia, muscle atrophy, exercise, exosome, muscle wasting, sarcopenia

## Abstract

Currently, no commercially available drugs have the ability to reverse cachexia or counteract muscle wasting and the loss of lean mass. Here, we report the methodology used to develop Physiactisome—a conditioned medium released by heat shock protein 60 (Hsp60)—overexpressing C2C12 cell lines enriched with small and large extracellular vesicles. We also present evidence supporting its use in the treatment of cachexia. Briefly, we obtain a nanovesicle-based secretion by genetically modifying C2C12 cell lines with an *Hsp60*-overexpressing plasmid. The secretion is used to treat naïve C2C12 cell lines. Physiactisome activates the expression of PGC-1α isoform 1, which is directly involved in mitochondrial biogenesis and muscle atrophy suppression, in naïve C2C12 cell lines. Proteomic analyses show Hsp60 localisation inside isolated nanovesicles and the localisation of several apocrine and merocrine molecules, with potential benefits for severe forms of muscle atrophy. Considering that Physiactisome can be easily obtained following tissue biopsy and can be applied to autologous muscle stem cells, we propose a potential nanovesicle-based anti-cachexia drug that could mimic the beneficial effects of exercise. Thus, Physiactisome may improve patient survival and quality of life. Furthermore, the method used to add Hsp60 into nanovesicles can be used to deliver other drugs or active proteins to vesicles.

## 1. Introduction

Cachexia, sarcopenia, and anorexia are all characterised by muscle wasting caused by disuse or an underlying disease. Muscle wasting and the consequent decrease in lean body mass affect the quality of life of patients by limiting mobility and can interfere with the efficacy of therapeutics. Although the benefits of physical activity in the context of these conditions are well known, prescribing exercise for these pathological conditions is not a simple undertaking, and alternative approaches to mimic the effects of exercise are needed [1].

Various acute or chronic medical conditions that result in hospitalisation and a decrease in patient activity result in cachexia, including cancer, amyotrophic lateral sclerosis, multiple sclerosis, muscular dystrophy, neuropathy, polio, spinal muscular atrophy, acquired immunodeficiency syndrome, congestive heart failure, chronic obstructive pulmonary disease, renal failure, and other conditions that result in chronic inflammation [1].

Sarcopenia is generally age-related and is characterised by a decrease in skeletal muscle mass due to decreased physical activity and the production of anabolic hormones [2]. Anorexia results from decreased energy intake, often due to a loss of appetite. However, malnutrition may also be observed in cachexia, even though cachexia is a disturbed metabolic state and cannot be treated by nutritional supplementation alone [1,3]. Cachexia, sarcopenia, and anorexia can also lead to cardiac muscle wasting, further emphasising the importance of managing muscle wasting in these conditions [4].

Skeletal muscle wasting is characterised by decreased protein synthesis in the skeletal muscle and the increased catabolism of skeletal muscle proteins, thereby resulting in sarcomere disintegration. In cachexia and sarcopenia, the release of reactive oxygen species (ROS) and reactive nitrogen species (RNS) by mitochondria contribute to muscle wasting. These interact to produce other highly reactive products (peroxynitrite and hydroxyl radical) that degrade intracellular structures, including myofibrils. When the production of radicals exceeds the buffering capacity of cells, oxidative stress and inflammation are induced. In C2C12 cells, ROS are known to upregulate the expression of E3 ubiquitin ligases [5], thereby increasing proteasomal activity and myofibrillar protein degradation.

The ubiquitin–proteasome system is the major proteolytic machinery that is systematically activated in cachexia [6]. A significant association has also been found with embryonic ectoderm development (EED) and histone deacetylase 1 (Hdac1), two important epigenetic regulators. Recent evidence suggests a mechanistic link between the aberrant acetylation/expression of transcription factors and wasting in diseased muscles through the dysregulated expression of cachexia-associated genes. *EED* encodes an evolutionarily conserved polycomb group protein that forms the core component of the polycomb repressive complex-2 (PRC2). The PRC2 complex plays an essential role in regulating chromatin structure and acts as an important regulator of cell development and differentiation during embryogenesis. HDAC1 is a component of the histone deacetylase complex. Histone acetyltransferases and deacetylases are critical enzymes that affect the expression of a variety of genes related to muscle wasting by modulating satellite cell activation and differentiation [7].

Currently, few drugs or molecules are known to counteract the muscle damage and wasting characteristic of cachexia. JMV2894 and hexarelin are able to counteract the mitochondrial impairment of muscle tissue in cisplatin-induced cachexia, with hexarelin being particularly effective on PGC-1α expression [8]. Ginsenoside Rg3 is also able to increase PGC-1α expression in DEX-induced myotube atrophy, an in vitro condition resembling cachexia [9]. Despite the promising results obtained with many drugs in preclinical models, only anamorelin, an orally active, selective high-affinity agonist of the ghrelin receptor, is currently in phase 3 trials for the treatment of cachexia. As the ghrelin receptor has anabolic and appetite-stimulating effects, anamorelin primarily serves as an anti-anorexia drug [10].

Currently, there are 16 molecules other than anamorelin being tested for the treatment of cachexia. However, all these treatments lag behind anamorelin in clinical trials; therefore, a cure for cachexia has not been approved to date. 

The most promising strategy to promote lean mass gain in cachexia is resistance training. Moreover, several clinical studies suggest that exercise slows the progression of various forms of cancer and increases the survival of cancer patients by reducing the risk of recurrence [11]. Exercise training decreases the expression of proinflammatory factors, including myostatin [12], tumour necrosis factor, and interleukin (IL)-6 [13,14] in skeletal muscle cells. Moreover, resistance training induces mTOR phosphorylation, a phenomenon that promotes protein synthesis in skeletal muscle [15]. Therefore, regular physical activity can reduce proinflammatory factors, improve the quality of life of patients, and positively influence their prognosis.

Physical activity and the regular contraction of skeletal muscle increase the production of ROS and RNS [16]. However, training also stimulates the production of antioxidant enzymes, such as manganese-dependent superoxide dismutase [17]. Furthermore, transcription factors such as PGC-1α are known to promote mitochondrial biogenesis and reduce muscle wasting during physical exercise [18]. This is especially important because it is known that the expression levels of PGC-1α-isoform 1, heat shock protein 60 (Hsp60), and IL-6 [19], which are all involved in mitochondrial biogenesis, are upregulated in soleus skeletal muscle cell lines derived from male mice after a single bout of endurance training. At the tissue level, resistance training improves muscle mass and strength by increasing the cross-sectional area of muscle fibres and the number of myofibrils [20].

Hsp60 is a molecule that is constitutively expressed in muscle cells, and its expression levels are known to be proportional to the mitochondria content and oxidative capacity of each muscle cell [21]. Hsp60 expression levels increase with training; however, this does not occur in all muscles. For example, Mattson et al. revealed a significant increase in Hsp60 levels in plantar muscle, and no difference in soleus muscle, in rats trained using an 8-week endurance protocol [22]. Morton et al. conducted a human study following endurance training and showed that Hsp60 levels in the lateral vastus muscle were significantly higher (25%) in trained individuals than in sedentary individuals [23]. Increased Hsp60 levels in trained mice facilitate mitochondrial protein import and folding, thereby inducing mitochondrial biogenesis [24].

In accordance with these findings, we have also reported an increase in Hsp60 levels in the fibres of the soleus muscle with training. However, we found a massive release of Hsp60-bearing exosomes in the blood of BALB/c mice after they completed a 6-week progressive endurance workout programme compared to sedentary animals [25]. We also attempted to isolate Hsp60-bearing small extracellular nanovesicles from the serum of endurance-trained mice for electron microscopy and Western blotting analyses; however, unfortunately, the number of vesicles was too low for further analyses. Our findings agree with previous findings that show that training increases the production of cytoprotective proteins, such as HSPs, and that Hsp60 may also be detected in exosomes, thereby demonstrating that it is not exclusively a mitochondrial protein [26].

The aims of the present study were to construct these Hsp60-bearing extracellular nanovesicles in vitro and release them into the bloodstream of trained animals in future experiments. We hypothesised that the nanovesicles are mediators of the beneficial effects of training, and we performed functional analyses of these nanovesicles to test our hypothesis. To achieve our objective of producing extracellular nanovesicles (small and large extracellular vesicles; EVs) rich in Hsp60, similar to those that we detected in the serum of endurance-trained mice [27], we devised an approach to obtain a conditioned medium enriched with both small and large EVs (Physiactisome), capable of activating the expression of PGC-1α-isoform 1 in naïve C2C12 cell lines and mouse immortalised myoblasts, which are widely used to study skeletal muscle homeostasis. Our EVs are naturally secreted, biolipid-based, cell-derived nanovesicles, which should be more biocompatible and host-friendly compared to artificial nanoparticle-based nanocarriers [28]. Although the literature is full of articles discussing the nomenclature of extracellular vesicles, such as the recent revision of their historical nomenclature [29], we chose to define our vesicles as nanovesicles because we, as morphologists, recognise them as nanosized vesicles. Based on the population of vesicles found with a Zetasizer, we then distinguished nanovesicles into small extracellular vesicles if they were between 80 and 200 nm in size and large extracellular vesicles if they were larger than 200 nm.

PGC-1α is directly related to mitochondrial biogenesis and the suppression of muscle atrophy, and optimal mitochondrial function contributes to muscle mass maintenance. Hence, this strategy is relevant to the biotechnology and pharmaceutical sectors, as it provides a process for the in vitro preparation of vesicles containing the thermal shock protein Hsp60, which can be used to prepare pharmaceutical compositions for the treatment of cachexia or muscle damage.

## 2. Materials and Methods

### 2.1. Cell Culture

C2C12 cell lines (mouse C3H muscle myoblasts, Cat. no. 91031101, ATCC: CRL1772, Sigma-Aldrich–Merck, Saint Louis, MO, USA) were grown following the seller’s instructions. Details on the cell culture method are described in the Appendix A.

### 2.2. Plasmid and Bacterial Transformation

The plasmid pCMV-6-Entry vector containing *HSPD1* (pCMV-6-Entry-HSPD1, Cat. no. MR222671, OriGene, Rockville, MD, USA) was used to overexpress the protein Hsp60. The sequence of the entire plasmid and details on the plasmid preparation are shown in the Appendix A. 

### 2.3. Preparation of Physiactisome

Physiactisome is obtained by inducing *HSPD1* overexpression in C2C12 cell lines, a condition that mimics Hsp60 overexpression in the skeletal muscles of mice following endurance training. 

In brief, the invention consists of a conditioned culture medium of immortalised myoblast cell lines, which overexpress the sequence coding for the heat shock protein of 60 kDa and isolate vesicles containing the heat shock protein of 60 kDa Hsp60 (Italian patent N. 102018000009235—9 September 2020; International application number—PCT/IB2019/058337) from the conditioned culture medium.

Details on the Physiactisome preparation method can be found in the Appendix A.

Physiactisome is the first patented EV-based drug against cachexia that could mimic physical exercise and counteract muscle wasting and the loss of lean body mass.

### 2.4. Treatment with Physiactisome

After 72 h from transfection, 2 mL medium was obtained from each well of a 6-well plate containing the engineered cell line at a density of 50,000 cells per well. Half of the medium (1 mL) was diluted to 1:2 in Dulbecco’s phosphate-buffered saline, and the other 1 mL was added to the serum-free medium in 6-well plates containing the naïve C2C12 cell line at a density of 50,000 cells per well to assess the effect of the factors released into the medium. Given the small volume of the medium, it was not possible to isolate small and large EVs by ultracentrifugation. RNA was extracted from the cell line 6 h after the beginning of the treatment to perform real-time polymerase chain reaction (qPCR). Human recombinant Hsp60 produced in our laboratory was used as a control in the naïve C2C12 treatment and qPCR experiments.

### 2.5. Isolation of Total RNA 

Total RNA from the conditioned medium-treated cell line was extracted using Tri Reagent (Cat. no. 93289, Sigma-Aldrich–Merck) according to the manufacturer’s instructions. RNA concentrations were determined using a NanoDrop ND-2000 (Thermo Fisher Scientific, Waltham, MA, USA). 

### 2.6. Quantitative Reverse Transcription Polymerase Chain Reaction

Reverse transcription was performed using an ImProm-II Reverse Transcriptase Kit (A3800, Promega, Madison, WI, USA) according to the manufacturer’s instructions. The qPCR analysis was performed using GoTaq qPCR Master Mix (A6001, Promega). mRNA levels were normalised to that of *GAPDH*. Changes at the transcript level were calculated using the 2^–ΔΔCT^ method [30]. Complementary DNA (cDNA) was amplified using the primers shown in Table 1; cDNA was amplified using a Rotor-gene™ 6000 real-time PCR machine (Qiagen, Hilden, Germany). The pair of primers named PGC1 αtot recognised all the isoforms of *PGC1α* because they amplify a 150 bp fragment of the domain common to all known isoforms.

### 2.7. Transmission Electron Microscopy in 3D Cultures

A three-dimensional culture in collagen gel was used to visualise small and large EVs because this type of method allows for a non-invasive assessment of whether EVs are released from cells. For details, see Appendix A.

### 2.8. Isolation of C2C12 EVs

To isolate small and large EVs from the myoblast-conditioned media for Western blot analysis, a cell line was cultured in T162 flasks (5000 cells cm^2^. When the cell line was at approximately 70–80% confluence, the medium was replaced with serum-free Dulbecco’s modified Eagle medium, and the cell line was starved for approximately 20 h. Following this, the medium was collected for the purification of EVs. For details, see Appendix A.

### 2.9. Size Distribution of C2C12 EVs

EV size distribution was measured by employing a dynamic light scattering system using a Zetasizer Nano S (Malvern Instruments, Malvern, UK). Briefly, 40 µL of undiluted extracted pellet in phosphate-buffered saline (PBS) was analysed at 25 °C by using a precision quartz cell for low volumes (ZEN2112). The refractive index and viscosity of the PBS dispersant were 1340 and 11,000 cP, respectively, at 25 °C [32].

### 2.10. Western Blotting

Cell lines and EVs were lysed in RIPA buffer (150 mM NaCl, 0.5% sodium deoxycholate (Sigma-Aldrich–Merck, St. Louis, MO, USA), 1% Triton X-100 (Sigma-Aldrich–Merck), 50 mM Tris, pH 8.0, 0.1% sodium dodecyl sulphate, cocktails of protease inhibitors (cOmplete ULTRA Tablets, Mini, EDTA-free, EASYpack, Sigma-Aldrich–Merck)). 

The following antibodies were used: anti-Alix (1A12, sc-53540, Santa Cruz Biotechnology; diluted 1:1000), anti-calnexin (MA3-027, Thermo Fisher Scientific; diluted 1:500), anti-beta-actin (AC-74, Sigma-Aldrich–Merck; diluted 1:5000), anti-Hsp70 (cmHsp70.1, produced in laboratory by Prof. Gabriele Multhoff [33]; diluted 1:1000), anti-Hsp60 (mouse monoclonal antibody ab13532, Abcam, Cambridge, UK), and anti-Rab5 (R4654, Sigma-Aldrich–Merck; diluted 1:500), all diluted in 5% milk in T-TBS. HRP-conjugated polyclonal rabbit anti-mouse immunoglobulin (P026002-2, Agilent–DAKO, Santa Clara, CA, USA) and HRP-conjugated polyclonal swine anti-rabbit immunoglobulin (P021702-2, Agilent–DAKO) diluted 1:2000 in 5% milk in T-TBS were used as secondary antibodies. Immunoreactive protein signals were detected using a Pierce ECL Western Blotting Substrate (Cat. no. 32106, Thermo Fisher Scientific) and captured using a ChemiDoc Imaging System (Bio-Rad, Hercules, CA, USA).

### 2.11. Recombinant Hsp60

See Appendix A. 

### 2.12. Proteomic Analysis of Small and Large EVs

Aliquots of 30 µg of small and large EV extracts were first dialysed, lyophilised, and then purified with the PlusOne 2-D Clean-Up Kit (GE Healthcare Life Sciences) according to the manufacturer’s recommendations. The pellets obtained were suspended in 100 μL of 50 mM ammonium bicarbonate (pH 8.3) and then incubated on ice for 15 min. Following this, 100 μL of 0.2% RapiGest SF (Waters, Milford, MA, USA) in 50 mM ammonium bicarbonate (pH 8.3) was added to the pellet and it was incubated on ice for 30 min. Protein concentration was determined using the Qubit Protein Assay kit and the Qubit 1.0 Fluorometer (Thermo Fisher Scientific).

The small EV sample had a concentration of 0.33 μg/μL, and the large EV sample had a protein concentration of 0.47 μg/μL. A total of 0.2 µg of yeast enolase was added as an internal standard to each sample. Proteins were reduced by adding 20.35 μg (small EV sample) and 29.3 μg (large EV sample) of DTT dissolved in 50 mM ammonium bicarbonate (pH 8.3). The solutions were kept in the dark for 3 h at 25 °C. Subsequently, alkylation was performed by the addition of iodoacetamide at the same molar ratio over total thiol groups and the reaction allowed to proceed for 1 h in the dark at 25 °C. 

Finally, the reduced and alkylated proteins were subjected to digestion using modified porcine trypsin (Promega) in ammonium bicarbonate (pH 8.3) at an enzyme-substrate ratio of 1:50 (37 °C overnight) [34,35]. The protein digests were dried under a vacuum and redissolved in 40 μL of 5% FA (final concentration 250 ng/μL).

### 2.13. Liquid Chromatography and Tandem Mass Spectrometry

Mass spectrometry (MS) data were acquired using an Orbitrap Fusion Tribrid (Q-OT-qIT) mass spectrometer (Thermo Fisher Scientific) equipped with a Dionex UltiMate 3000 RSLCnano system (Thermo Fisher Scientific). 

MS calibration was performed using the Pierce LTQ Velos ESI Positive Ion Calibration Solution (Thermo Fisher Scientific). MS data acquisition was performed using Xcalibur v. 3.0.63 software (Thermo Fisher Scientific).

For details, see Appendix A.

### 2.14. Database Searches, Protein Identification, and Label-Free Quantification Analysis

LC–MS/MS data were processed using PEAKS software v. X (Bioinformatics Solutions, Waterloo, ON, Canada). The data were searched against the 17,449 “Mus musculus” Swiss-Prot database (release February 2019), to which the yeast enolase 1 (P00924) sequence was added. Tryptic peptides with a maximum of three missed cleavage sites were subjected to an in silico search. See Appendix A for details.

### 2.15. Interaction Analysis

FunRich (functional enrichment analysis tool) is a new tool (FunRich. Available online: http://www.funrich.org—accessed on 1 May 2019) mainly used for functional enrichment and interaction network analysis of genes and proteins [36]. We applied FunRich to analyse the interaction networks of the proteins identified in large and small EVs.

### 2.16. Statistical Analysis

The obtained results were statistically analysed using ANOVA with Bonferroni post-test. All statistical analyses were performed using GraphPad Prism TM 4.0 software (GraphPad Software, San Diego, CA, USA). All data are presented as means ± SD, with the level of statistical significance set at *p* < 0.05.

## 3. Results

### 3.1. C2C12 Cell Lines Secrete Nanovesicles

Electron microscopy analysis of differentiating C2C12 cell lines revealed the presence of EVs with diameters ranging from 50 to 140 nm next to the cell membranes (Figure 1A); these findings are similar to previous observations [37]. Additionally, immunogold analyses revealed these EVs to be dispersed throughout the extracellular matrix and to contain the exosome marker Alix (Figure 1B,C), indicating sub-membranous localisation.

The culture medium collected after a 24 h incubation of the cell line in serum-free medium was found to contain a population of large EVs (peak 1 at 201.4 nm and peak 2 at 1256 nm in diameter, isolate at 15,000× *g*; Figure 2A) and small EVs (peak 1 at 123 nm in diameter, isolated at 110,000× *g*; Figure 2B).

Western blotting of the two EV fractions revealed that the small EVs were significantly enriched in classical exosome (small EV) proteins, such as Alix, Hsp70, and RAB5 (Figure 3); the large EVs contained RAB5 and Hsp70 but not Alix (Figure 3). The negligible levels of Alix and RAB5, as well as the presence of calnexin, in the total cell lysate confirmed the high quality of the preparations. Alix is a typical protein usually found in small EVs and to a lesser extent in large EV fractions [38], while RAB5 is a GTPase considered an early endosome marker, which plays an important role in membrane transport [39]. These findings suggest that our separation method is optimal (Figure 3). These results also demonstrated that C2C12 cell lines are capable of secreting both small and large EVs and that our isolation method allowed for the recovery of both fractions without contamination by cell fragments. Hsp60 was not detected in the lysates of the small EVs of the control samples, as reported previously (Appendix A) [26].

### 3.2. Proteomic Analysis of Small and Large EVs

A total of 1545 and 1385 proteins were identified in the large and small EVs, respectively (Appendix A); this included ENO1, which was used as an internal standard to quantify relative protein abundance.

In terms of relative abundance, a label-free comparative approach revealed that 395 proteins exhibited differences in expression levels of more than twofold, with a *p* value of 0.05 (*t*-test); these included 168 upregulated and 227 downregulated proteins in small EVs compared to those in large EVs. In total, 569 proteins were common in both samples, with differences in relative abundance of less than twofold; 421 proteins were exclusively present in small EVs, and 581 were exclusively present in large EVs (Figure 4A,B and Appendix A).

The proteomic data were consistent with the Western blot results regarding the expression of small EV markers, such as Alix (UniProt accession no., Q9WU78), which was identified among the differentially accumulated proteins. Lower Alix levels were observed in large EVs than in small EVs (0.12-fold change, Appendix A).

Both Hsp70 (UniProt accession no. P17879 and P16627) and RAB5 (UniProt accession no. P35278 and Q9CQD1) were identified in small and large EVs (with no differences in relative abundances). Calnexin was identified as a differentially expressed protein having a lower abundance in small EVs than in large EVs. Moreover, APOA1/2 and albumin generally coisolated with EV structures as contaminants were absent, suggesting the good quality of our preparations. Interestingly, among the list of the top 100 proteins commonly identified in exosomes (data derived from ExoCarta), 44 proteins were also identified in our experiments.

To investigate the overall proteomic composition of large and small EVs, the identified proteins were categorised using FunRich and subjected to a gene interaction network analysis (Figure 5A,B). Proteins identified in small and large EVs were significantly associated with ubiquitin (Ubc), EED, E3 ubiquitin ligase MIB1 (Mib), and forkhead box protein P3 proteins. Significant associations with protein/nucleic acid deglycase DJ-1 (Park7), RNA-binding protein EWS (Ewsr1), and gap junction alpha-1 protein (Gja1) were found for proteins identified in large EVs. In contrast, a significant association with Hdac1 was found for proteins identified in small EVs.

### 3.3. Development and Characterisation of Physiactisome

C2C12 cell lines were transfected with either pCMV-6-Entry-*HSPD1* plasmid, which encodes the Hsp60 var1 mouse gene (Physiactisome), or with the pCMV-6-Entry plasmid (Control). Transfection efficiency was monitored using flow cytometry (data not shown) to assess the number of Myc-DDK-positive cell lines, which accounted for 7–8% of the total cell population.

A confocal analysis confirmed the presence of the cell line expressing high levels of Hsp60. Cell lines transfected with pCMV-6-Entry-*HSPD1* exhibited foci (Appendix A).

### 3.4. Effect of Physiactisome on the Expression of PGC-1α Isoform 1

Physiactisome was able to significantly induce the expression of total *PGC-1α*, as well as that of the *PGC-1α isoform 1* at the transcript level (Figure 6A). This experiment suggested that Hsp60 was responsible for the observed effects of Physiactisome, as it contained high levels of Hsp60.

To investigate whether the overexpressed Hsp60 was indeed responsible for the upregulation of PGC-1α, we treated naïve and undifferentiated C2C12 cell lines with 20 and 40 ng mL^−1^ of recombinant human Hsp60 synthesised in our laboratory (hrHsp60, sequence homology between human and mouse = 90%). Transcript level analysis of the expression of *PGC-1α* and its isoform 1 revealed that hrHsp60 could induce the expression of *PGC-1α* and that of the specific *PGC-1α isoform 1* in a dose-dependent manner, although the data are not statistically significant with *p* > 0.05 (Figure 6B).

Considering that the hrHsp60 synthesised in our laboratory was designed without a mitochondrial localisation sequence, this experiment also demonstrated that the effect elicited by Hsp60 on *PGC-1α* transcript levels does not require mitochondrial localisation. Interestingly, Hsp60 was identified among the proteins that were differentially expressed between small and large EVs (Figure 7), suggesting that the nanovesicles secreted by C2C12 cell lines contain Hsp60. In particular, Hsp60 (UniProt Acc. P63038) was identified as having a higher abundance in large EVs than in small EVs (Figure 7B, 25-fold change, Appendix A). This abundance is also shown in Appendix A.

Taken together, these data show that, among the hundred proteins present within the small and large EVs secreted by C2C12 cell lines, Hsp60 plays a major role in upregulating the expression of *PGC-1α*.

## 4. Discussion

In the present study, we described the methodology for the production of Physiactisome, which is a conditioned medium released by Hsp60-overexpressing C2C12 cell lines that contain high levels of Hsp60 and small and large EVs and that are able to activate the expression of *PGC-1α isoform 1*. This product can be used for the treatment of cachexia and muscle atrophy, as PGC-1α is involved in mitochondria biogenesis and is able to counteract muscle wasting. We demonstrated that C2C12 cell lines release large EVs into the culture medium in addition to small EVs, which contain diverse proteomes, including Hsp60, as previously described by our research group (Figure 1 and Figure 2) [27]. The release of small active EVs by C2C12 cell lines has also been previously demonstrated by Guescini et al. [37]; they demonstrated the release of small EVs with an average diameter of 90–99 nm in the culture medium of normal and hydrogen peroxide-treated C2C12 cell lines. Their experiments suggest an anti-myogenic differentiation and proliferative effect of the medium containing C2C12-derived small EVs.

Label-free quantitative proteomic analyses allowed us to determine the proteomes of small and large EVs. Hsp60 was 25 times more abundant in large EVs than in small EVs. Interaction network analyses showed that the proteins present in both small and large EVs were significantly associated with Ubc and Mib. Recently, it has been suggested that the ubiquitin–proteasome pathway may regulate epigenetic changes [40]. These findings suggest that small and large EVs may deliver factors such as the members of the proteasome pathway, which are also involved in epigenetic regulation.

We also demonstrated that Physiactisome can activate the transcription of *PGC-1α*, particularly the *PGC-1α isoform 1*. Specifically, we collected the medium from cell lines transfected with the pCMV-6-Entry-*HSPD1* plasmid or a negative control empty plasmid and used it to treat un-transfected C2C12 cell lines. As a positive control, we used cell lines treated with 20 and 40 ng/mL hrHsp60. We evaluated the expression of all the different isoforms of *PGC1α*, with special focus on the α1 isoform. Both Physiactisome and the 40 ng mL^−1^ hrHsp60 induced the expression of total *PGC-1α* and *PGC-1α isoform 1* (Figure 6). These experiments demonstrated the following: (a) Physiactisome activates the transcription of *PGC-1α isoform 1*; (b) the likely bioactive molecule in Physiactisome is Hsp60; (c) the Myc-DDK tag does not affect Hsp60 activity, as hrHsp60 (which contains a poly-histidine tag (His-tag) instead of the Myc-DDK tag) exhibits effects comparable to those of Physiactisome; (d) the domain that activates the expression of *PGC-1α* is located in the central region of Hsp60 and does not overlap with the mitochondrial localisation sequence; (e) the activity of Hsp60 is not species-specific, as Hsp60 from humans and mice activate the transcription of the same genes; and (f) Hsp60 contained in EVs is more active than 40 ng mL^−1^ hrHsp60 (freely present in the medium) because this level is 3–10 times higher than that detected in the serum of trained animals and could also increase the expression of the different isoforms of *PGC-1α* [26].

Physiactisome, an exercise mimic, is the only patented drug for the treatment of cachexia and sarcopenia based on extracellular nanovesicles. One of its limitations is that the molecular method for the overexpression of Hsp60 and EV purification must be further improved to obtain stable transfection and Physiactisome-producing C2C12 cell lines to scale up production. Methods to isolate an EV subpopulation are subjects of constant debate and, although ultracentrifugation is the most commonly used, other techniques, such as size exclusion chromatography (SEC) using different matrices (e.g., Sepharose 2B, Sephacryl S-400), filtration, or precipitation, have been utilised [41]. More recently, a combination of different methods (e.g., ultracentrifugation and SEC) are being employed to improve the purity of EVs [42]. A second limitation of this type of drug is the adverse response of the immune system. However, the experimental method used to obtain Hsp60-bearing extracellular nanovesicles may be applied to autologous muscle stem cells isolated from patient biopsies. Hence, a stable autologous muscle cell line may be established for each patient and can be scaled up to production in a GMP facility. Industrial ultracentrifuges are now commercially available for the continuous isolation of nanovesicles. An autologous and personalised drug will allow the patient to overcome rejection and an adverse immune response. A third limitation may be the administration of a nanovesicle-based drug. In this case, an initial experiment should be performed on cachectic animals by injecting Physiactisome directly in the tibialis anterior muscle since an injection would be better tolerated, allowing for an understanding of the fate of Physiactisome and its effects on the damaged muscle. It may be difficult and time consuming to bring this product to market, but it will be the future of personalised drugs, as natural nanovesicles may overcome many of the problems of promising nanoparticles. Moreover, the method used to add Hsp60 to the nanovesicles may be used to deliver other drugs or active proteins to the vesicles. This method may be preferable to the direct manipulation of extracellular vesicles because it preserves the membrane lipid composition and protein content of vesicles released from muscle cells. Although attempts can be made to obtain manipulated vesicles from cells other than muscle cells (e.g., cancer cells), we believe that, for proper uptake of the manipulated vesicles from the target cell, it is important to choose the same type of cell as the target cell. This can be demonstrated by extracting green vesicles from GFP-expressing cells and treating naïve cells.

In conclusion, our study suggests that Physiactisome treatment of C2C12 cells recapitulates the conditions prevalent during the muscle response in trained mice [26]. Production scale-up of Physiactisome by employing a strategy for a more stable overexpression of Hsp60 and improving the purification of ultrapure vesicles would allow the production of large quantities of the drug. 

Physiactisome, by enhancing the expression of *PGC-1α* in cachectic muscle, may reduce ROS generation and increase mitochondrial biogenesis, thus favouring a good balance between fusion and fission protein activities. Moreover, the overexpression of PGC-1α in vivo suppresses immobilisation-induced autophagy in mitochondria (termed mitophagy), a condition reported to play an important role in cancer cachexia [43,44]. Mitochondria are responsible for the regulation of total protein turnover of the skeletal muscle; thus, their dysfunction results in skeletal muscle wasting and a decrease in lean body mass typical of cancer cachexia. Drugs that can target mitochondrial biogenesis are emerging as new strategies to combat cachexia, thus ameliorating this metabolic syndrome.

## 5. Patents

B.R., T.E., D.D., M.F., C.C., M.G.A., M.G., G.P., C.D., A.S., C.F. and D.F.V. declare that they have applied for a patent for the production of EVs (Italian Patent n. 102018000009235—08/10/2018, PCT/IB2019/058337—01/10/2019, INTERNATIONAL PUBLICATION N. WO20200750074 PUBLICATION DATE 16/04/2020).

## Figures and Tables

**Figure 1 cells-11-01406-f001:**
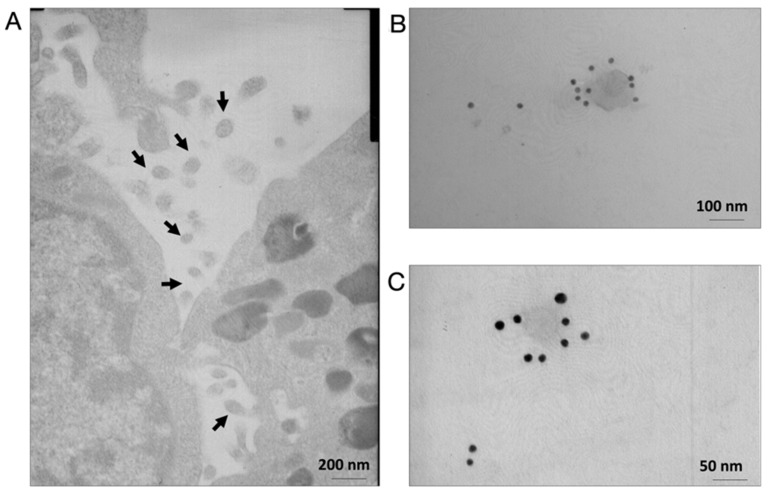
C2C12 cell lines secrete extracellular vesicles (EVs). (**A**) Electron microscopy analyses of 3D cultures of C2C12 cell lines show the presence of budding intermediates (nascent vesicles, black arrows) at the plasma membrane or already budded vesicles with an average diameter of 80 nm (black arrows). (**B**,**C**) Transmission electron microscopy showing immunogold labelling of Alix (a known marker of small EVs) in nanovesicles from the 3D cultures of C2C12 cell lines (10 nm gold particles).

**Figure 2 cells-11-01406-f002:**
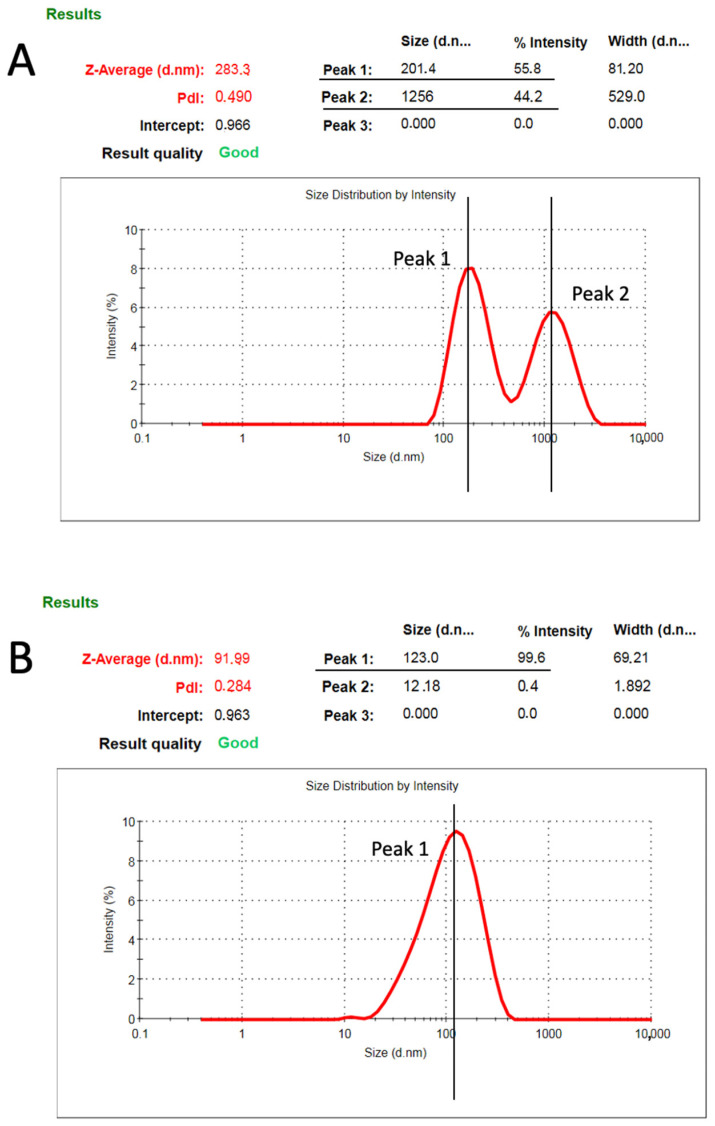
ζ-Potential measurements of particles in the C2C12 culture medium, following centrifugation at 15,000× *g* (large EVs, (**A**)) and at 110,000× *g* (small EVs, (**B**)).

**Figure 3 cells-11-01406-f003:**
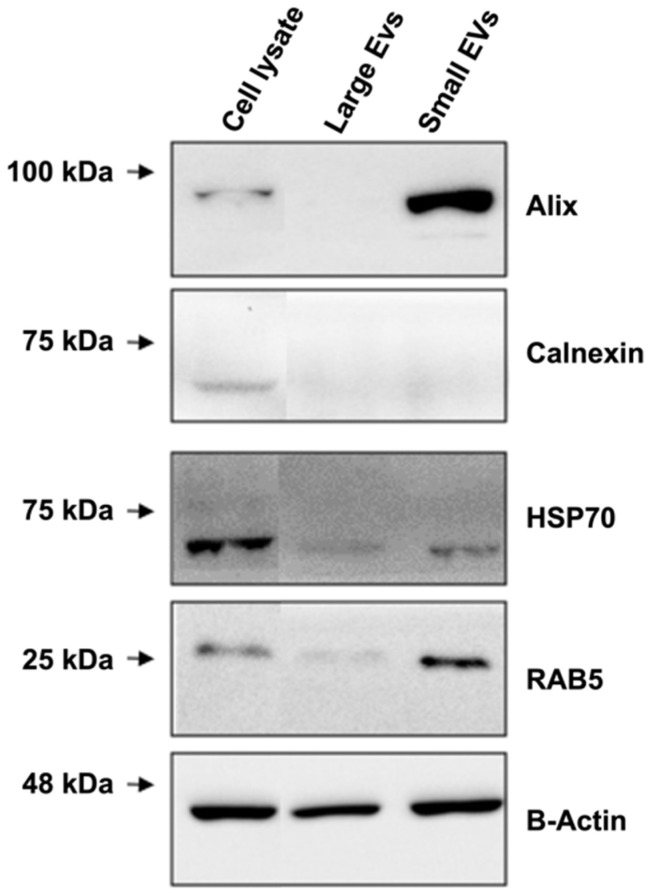
Western blot analysis of large extracellular vesicles (EVs), small EVs, and total lysates from C2C12 cell lines (10 μg of protein per lane) to detect the expression of Alix, calnexin, HSP70, and RAB5. Β-Actin was used as a loading control.

**Figure 4 cells-11-01406-f004:**
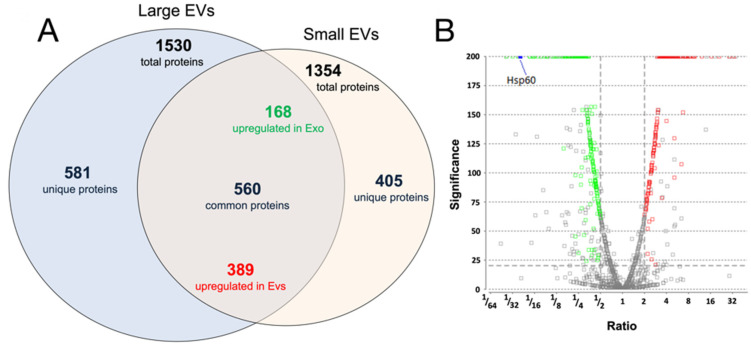
(**A**) Venn diagram showing the identified proteins (unique, common, and up-/down-regulated) between large extracellular vesicles (EVs) and small EVs. (**B**) Volcano plot of significant proteins differentially expressed between large EVs and small EVs. The X axis represents the fold change of differentially accumulated proteins represented with the ratio (large/small EVs), and the Y axis represents the corresponding significance. Fold change ≥ 2 and *t*-test with a *p* < 0.05 were set as the significance threshold for differential expression. Green represents downregulated proteins, and red represents upregulated proteins (large EVs/small EVs).

**Figure 5 cells-11-01406-f005:**
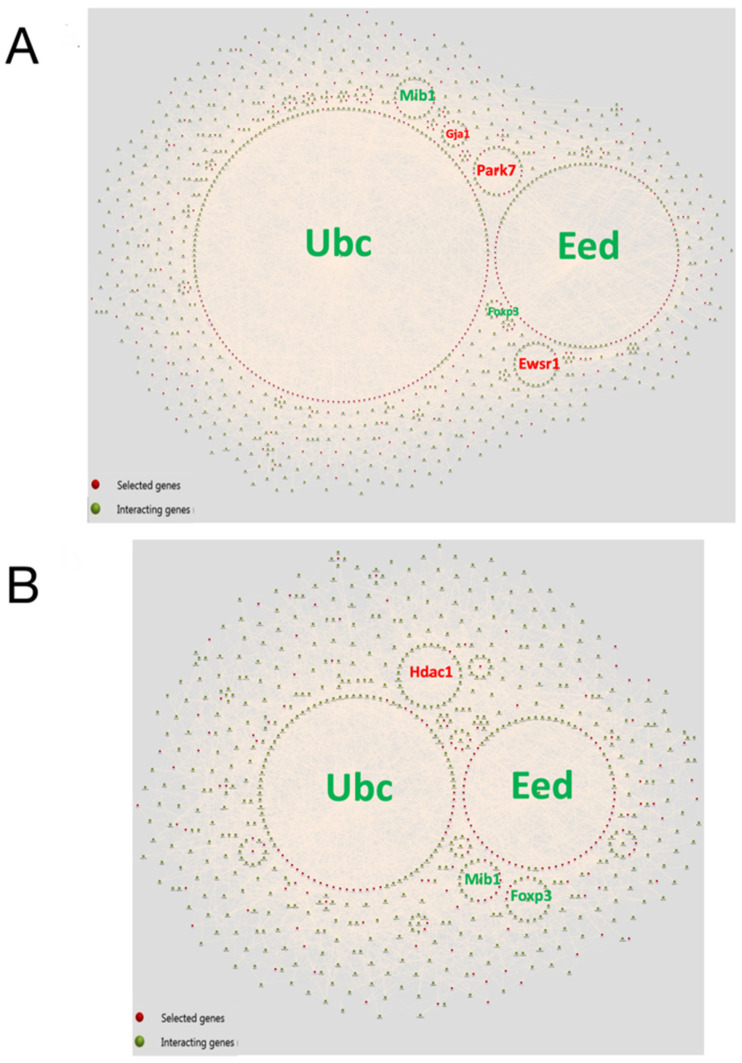
Interaction diagrams of proteins identified in large extracellular vesicles (EVs) and small EVs created using FunRich. The list of total proteins identified in large EVs (**A**) and small EVs (**B**) was uploaded, and interaction analysis tool was used to obtain the diagrams. The red and green spheres indicate the input proteins (selected genes) and the interacting proteins (interacting genes), respectively.

**Figure 6 cells-11-01406-f006:**
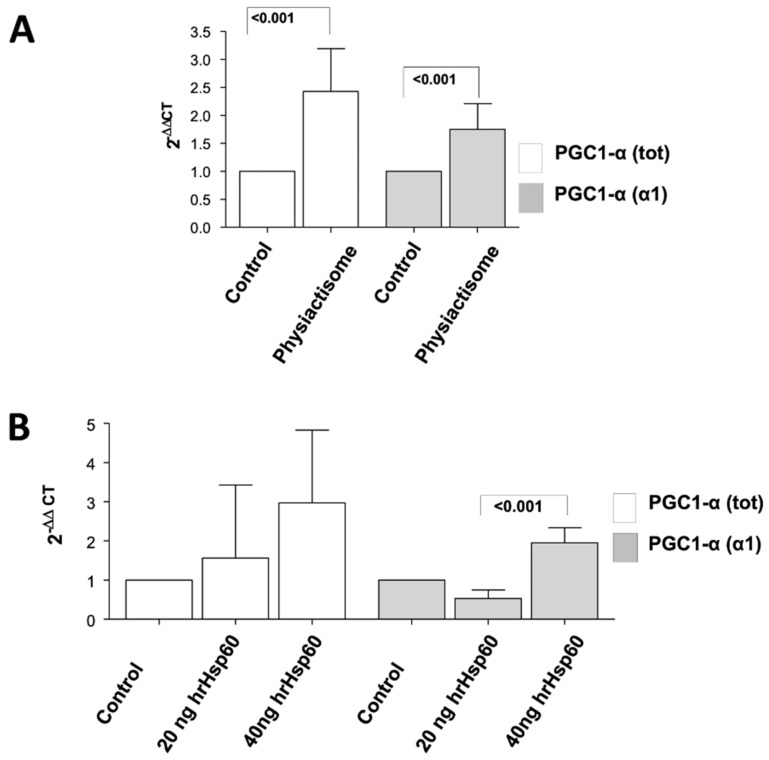
Bars show the transcript level expression of *PGC1α* isoforms (*PGC1α* total (α tot), isoform 1 (α 1)) as revealed by real-time PCR (values are normalised to those of the reference genes). Fold change was calculated according to the Livak Method (2^−ΔΔCT^) in C2C12 cell lines (naïve) treated with Physiactisome (**A**) or C2C12 cell lines (naïve) treated with recombinant human Hsp60 (hrHsp60, (**B**)).

**Figure 7 cells-11-01406-f007:**
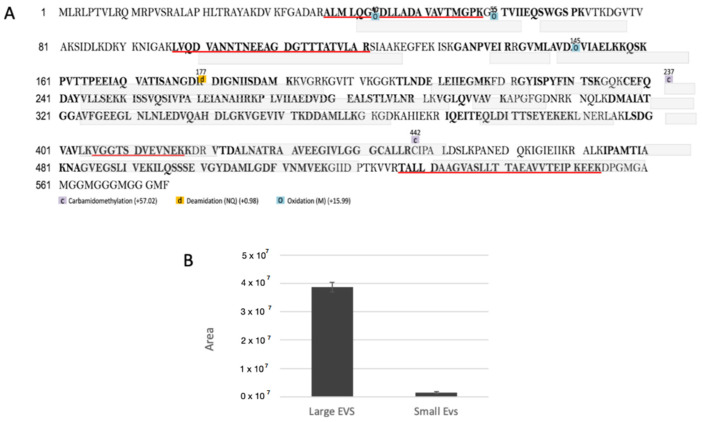
(**A**) Sequence coverage map of Hsp60 (UniProt accession no., P63038) obtained by tryptic digestion and nUHPLC/HR nESI-MS/MS. Peptides identified in the extracellular vesicle (EV) samples are shown in bold; peptides identified in the small EVs sample are underlined. (**B**) Quantification of Hsp60 levels between large and small EVs.

**Table 1 cells-11-01406-t001:** Forward and reverse primers used for qRT-PCR.

Primer	Target Sequence	Forward	Reverse
** *PGC1 tot* **	[31]	5′-TGATGTGAATGACTTGGATACAGACA-3′	5′-GCTCATTGTTGTACTGGTTGGATATG-3′
** *PGC1 α1* **	[31]	5′-GGACATGTGCAGCCAAGACTCT-3′	5′-CACTTCAATCCACCCAGAAAGCT-3′
** *GADPH* **	MGI:MGI:95640	5′-CAAGGACACTGAGCAAGAGA-3′	5′-GCCCCTCCTGTTATTATGGG-3′

## Data Availability

Data are contained within the article or Appendix A. The data not shown will be available upon request due to patent restrictions.

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
