# Peer review of "Physiactisome: A New Nanovesicle Drug Containing Heat Shock Protein 60 for Treating Muscle Wasting and Cachexia"

_cells, 2022, doi:10.3390/cells11091406_

Round 1

Reviewer 1 Report

In the manuscript by Di Felice et al, the authors describe the characterization of nanovesicles containing Hsp60 and suggest that these can be used to treat cachexia and sarcopenia. The manuscript is limited in scope, but does establish that these vesicles can be produced and elicit the effect of Hsp60 on the C2C12 cell line. Several aspects of the manuscript should be improved. 

  1. In line 61, the authors refer to naïve C2C12 cells and mouse immortalized myoblasts. It is assumed that the authors meant that C2C12 cells are a mouse immortalized myoblast cell line. Throughout the text, cell line should be used rather than cells.
  2. The use of Physiactsome is overstated throughout the manuscript, including the title. Line 205 states that “ Physiactisome is the first patented EV-based drug against cachexia that can mimic physical exercise and counteract muscle wasting and loss of lean body mass.” The authors show no evidence that Physiactsome mimics physical exercise (is a 2 fold upregulation of PGC-1 α the only effect of exercise?) or has any effect on muscle wasting in vivo. The only evidence they show is that Physiactsome upregulates PGC-1 α mRNA in the C2C12 cell line.
  3. Figure 2 should be modified for legibility. The table and figure axis legends must be enlarged. The large EV and small EV peaks are highly overlapping- the most predominant peak in the large EV pool appears to almost entirely overlap with the small EV peak. Are these two distinct pools or a gradient of vesicle size with a predominant size at 100 nM?
  4. The text supporting Figure 3 is confusing. The authors state that Alix is a classical exosome (small EV) component in line 349, but then state that Alix is found in small and large EVs (line 352). The calnexin result is used for a strong conclusion on the cleanliness of the preps, but the image is very light. The western data should be quantitated.  Cropping is noted in the HSP70, Alix and calnexin blots- the authors must clearly indicate if different blots were used for the comparisons. The authors state that HSP60 was not detected in EVs but show no evidence to confirm this conclusion.
  5. In Figure 6, no statistics are shown for PGC1-α (tot) in B. Were the data not significant? Given that these data are the only data in the manuscript that support the activity of Physiactsome, a western blot of PGC1 α would strengthen the manuscript. It is acknowledged that PGC1 α stimulates mitochondrial biogenesis, but another assay demonstrating enhanced mitochondrial biogenesis would strengthen the manuscript.    
  6. The authors should show the mined data that establish that Hsp60 has a higher abundance in large EVs and include a representation of these data in Figure 6. Referring the reader to Supporting Information Tables S1-S4 asks the reader to independently perform this analysis.

Author Response

In the manuscript by Di Felice et al, the authors describe the characterization of nanovesicles containing Hsp60 and suggest that these can be used to treat cachexia and sarcopenia. The manuscript is limited in scope, but does establish that these vesicles can be produced and elicit the effect of Hsp60 on the C2C12 cell line. Several aspects of the manuscript should be improved. 

  1. In line 61, the authors refer to naïve C2C12 cells and mouse immortalized myoblasts. It is assumed that the authors meant that C2C12 cells are a mouse immortalized myoblast cell line. Throughout the text, cell line should be used rather than cells.

Response: I have made the necessary changes and used the phrase “cell line” at all relevant instances in the manuscript.

  1. The use of Physiactsome is overstated throughout the manuscript, including the title. Line 205 states that “Physiactisome is the first patented EV-based drug against cachexia that can mimic physical exercise and counteract muscle wasting and loss of lean body mass.” The authors show no evidence that Physiactsome mimics physical exercise (is a 2 fold upregulation of PGC-1 α the only effect of exercise?) or has any effect on muscle wasting in vivo. The only evidence they show is that Physiactsome upregulates PGC-1 α mRNA in the C2C12 cell line.

Response: On page 4, line 196 of the file, where the changes were highlighted in red using the Word function, we changed the word "can" to "could" to indicate that the implications of the patented drug are hypothetical. The sentence on page 4 is a part of the documentation used to register and patent the product called "Physiactisome." In the article, the term "Physiactisome" has been used because it is now a patented product  (Italian patent N. 102018000009235 – 9 September 2020; International application number - PCT/IB2019/058337); therefore, we can refer to it by its registered name.

To date, we have only been able to perform in vitro experiments because the quantity of the product is limited; thus we cannot  perform in vivo experiments yet. However, the project and the experimentation on cachectic animals (the Balc / c model - tumor C26) have already been presented to and approved by the Ministry of Health, and animal model-based experiments will be conducted in a future study and published subsequently. Physiactisome is a way to mimic what we demonstrated in the Scientific Report article published in 2016(Barone R, Macaluso F, Sangiorgi C, Campanella C, Marino Gammazza A, Moresi V, Coletti D, Conway de Macario E, Macario AJ, Cappello F, Adamo S, Farina F, Zummo G, Di Felice V. Skeletal muscle Heat shock protein 60 increases after endurance training and induces peroxisome proliferator-activated receptor gamma coactivator 1 α1 expression. Sci Rep. 2016 Jan 27;6:19781. doi: 10.1038/srep19781. PMID: 26812922; PMCID: PMC4728392.) by our research group and which concerns animal testing.

  1. Figure 2 should be modified for legibility. The table and figure axis legends must be enlarged. The large EV and small EV peaks are highly overlapping- the most predominant peak in the large EV pool appears to almost entirely overlap with the small EV peak. Are these two distinct pools or a gradient of vesicle size with a predominant size at 100 nM?

Response: We have replaced Figure 2 with the image of another experiment similar to the one shown, and ensured that it is legible. We have enlarged the image so that the details are clearly visible. We have also highlighted the peaks in both samples in the text as well as in the graph. "A" represents large EVs and the two peaks are at 201 and 1256 nm, while "B" represents small EVs and the single peak is at 123 nm with an intensity of 99.6%. We have also changed the text of the results paragraph where the values of the peaks are indicated accordingly.

Page 8 line 340 – 343 changed to: “The culture medium collected after 24 h-incubation of the cells in serum-free medium was found to contain a population of large EVs (peak 1 at 201,4 nm and peak 2 at 1256 nm in diameter, isolate at 15,000 × g; Figure 2A) and small EVs (peak 1 at 123 nm in diameter, isolated at 110,000 × g; Figure 2B).”

  1. The text supporting Figure 3 is confusing. The authors state that Alix is a classical exosome (small EV) component in line 349, but then state that Alix is found in small and large EVs (line 352).

Response: Our intention was to highlight the differences in protein content between "small" and "large" EVs. We agree that this section was ambiguous and have separated the comments on Alix and RAB5 and supported the results with two very recent references. See page 8- 9 , from line 343 to 345.

The calnexin result is used for a strong conclusion on the cleanliness of the preps, but the image is very light. The western data should be quantitated.

All WBs in their original form have been included in the submission as supplementary material as requested by the journal. The lanes have been cut due to the need for space in the image. In response to the reviewer concerns we have decided to include other WBs as Supplementary material in their original and complete form.

Regarding the calnexin, as seen in the original images of calnexin WB, the nitrocellulose filter was cut into several pieces, considering the markers, to avoid stripping, and also to get understand the relative expression of calnexin compared with the other proteins expressed by the same vesicles and sample. Thus, calnexin is weakly expressed in our samples compared to actin and Alix. However, as suggested by the referee, we have included a graph quantifying calnexin relative to actin from the same filter and WB. Proteomic analysis confirmed the WB result. "Calnexin was identified as a differentially expressed protein having a lower abundance in small EVs than in large EVs. " see page 10, lines 387-388.

  1. C) Cropping is noted in the HSP70, Alix and calnexin blots- the authors must clearly indicate if different blots were used for the comparisons.

Yes, the filters were often trimmed, as mentioned earlier, to use more antibodies on the same WB without having to perform the stripping method or to arrange the samples differently than in the original WB. However, as several experiments were performed, we opted to include another WB regarding the expression of Alix, Rab5, Hsp70, Hsp60, and actin in the supplementary material (the nitrocellulose filter in its entirety before it was cut and the WB strips derived from it) as supplementary Figure S2.

  1. D) The authors state that HSP60 was not detected in EVs but show no evidence to confirm this conclusion.

We have included another Supplementary Figure (named S1) showing the expression of Hsp60 in cell lysate and large EVs. We have revised numbering of the Supplementary figures accordingly.

  1. A) In Figure 6, no statistics are shown for PGC1-α (tot) in B. Were the data not significant?

We opted to include a description of the data, even if it is not statistically significant. Therefore, we have added the phrase "although the data are not statistically significant" on page 12 line 438.

  1. B) Given that these data are the only data in the manuscript that support the activity of Physiactsome, a western blot of PGC1 α would strengthen the manuscript. It is acknowledged that PGC1 α stimulates mitochondrial biogenesis, but another assay demonstrating enhanced mitochondrial biogenesis would strengthen the manuscript.    

We have the WB data on the expression levels of PGC-1 alpha total on the total lysate of transfected c2C12. The data were previously published our article (Barone et al., 2016). We could not perform a WB of the total PGC-1 alpha levels on the lysate of the cells treated with the conditioned medium because of the low yield of the transfected cells. Transient transfection does not last long and unfortunately, after 15 days, treatment with G418 only induces proliferation of C2C12 cells that are resistant to G418, does not retain the plasmid, and does not overexpress Hsp60-DDK. Indeed, to obtain the exosomes, we require 175 different flasks, which we have not been able to obtain. With the little material obtained, we can only visualize the increase in PGC-1 alpha in real time. We are attempting to obtain a stable line of C2C12 cells overexpressing HSP60 so that we can not only perform WB on the treated cells but also inject the exosomes and the different vesicles into the anterior tibial muscle of Balb / c mice with and without C26 tumor (with and without cachexia). These experiments are underway and the project with animals has already been approved by the Italian Ministry of Health.These data will be published in a future study.

  1. A) The authors should show the mined data that establish that Hsp60 has a higher abundance in large EVs and include a representation of these data in Figure 6.

Since these data were requested by the reviewer in a previous point, we have included the reference to Supporting Information Figure S1 on page 12, line 447, which shows the abundance of Hsp60 in large EVs, as also demonstrated by the proteomics analysis.

  1. A) Referring the reader to Supporting Information Tables S1-S4 asks the reader to independently perform this analysis.

For a better understanding of the proteomic analysis, we added a histogram to Figure 7 (Figure 7B) showing the average values of the areas used for quantification of the identified proteins and, in particular, HSP60. Moreover, we have moved the sentence "In particular, Hsp60 (UniProt Acc. P63038) was identified as having a higher abundance in large EVs than in small EVs (Figure 7B, 25-fold change, Supporting Information Tables S1 - S4)." from page 13 to page 12, before the reference to the new WB of Figure S1. We have also revised the figure legend accordingly.

Reviewer 2 Report

Di Felice et al aim at producing extracellular vesicles (EV) from muscular cells carrying the chaperone protein Hsp60 for release in distant muscular cells to improve their metabolic function, and counteract diseases characterized by muscle atrophy such as cancer cachexia. To this end, they first isolated small and large EVs from C2C12 cells and performed proteomic analyses to compare their cargo. They then transfected C2C12 cells with either a control plasmid or a plasmid overexpressing the protein Hsp60. They then treated naïve C2C12 cells with conditioned media (CM) enriched in Hsp60 or not and observed an increased expression of PGC1a. Treatment of naïve C2C12 cell with human Hsp60 protein induced a similar increase in PGC1a expression. Authors therefore conclude that this approach would represent a new therapeutic tool to enhance PGC1a expression in cachectic muscle, thereby promoting mitochondria biogenesis and reducing autophagy.

The manuscript is well written and easy to follow. Although interesting, I believe the study is missing key experiments to be really convincing and to support this technique as a new potential therapeutic approach. Therefore, it would be mandatory in my view to strengthen the actual data by the following experiments:

1) Key control experiments are lacking: It is mandatory to show that authors can successfully enrich Hsp60 into CM from C2C12 cells transfected with a plasmid overexpressing Hsp60 in comparison to cells transfected with a control plasmid. Transfection efficiency is quite low (7-8%), so it is essential to show that this is sufficient to significantly enrich CM in Hsp60 protein levels. Authors show in Figures 1-3 that they can successfully isolate small and large EVs from C2C12 CM. Therefore, I suggest that they isolate EVs from CM of cells overexpressing Hsp60 or not, and perform a western blot to confirm enrichment into secreted EVs.

2) By treating naive C2C12 cells with CM from C2C12 cells overexpressing Hsp60 or not, authors concluded that this CM (named Physioactisome) can increase PGC1a mRNA expression and therefore may improve mitochondrial biogenesis and counteract muscle wasting for treatment of cachexia. However, this is not supported by a sufficient amount of data:

A) Increased PGC1a expression in response to Physioactisome (CM enriched in Hsp60 protein) has already been reported in a previous paper from the same group (Figure 8D from Barone et al, Sci Rep, 2016, doi 1038/srep19781) and is not a novel observation. Furthermore, in this paper from 2016, authors only observed an increase in PGC1a expression in target C2C12 cells after 6 hours but not after 12 hours. It seems therefore that the effect is transient and authors would actually need to show that this affect downstream targets of PGC1a to convince of the efficiency of this treatment.

B) Authors use CM from undifferentiated C2C12 cells, and not from myotubes and apparently treat undifferentiated C2C12 cells to look at PGC1a expression. This is, in my view a critical point. The use of differentiated C2C12 myotubes is essential to study the metabolism of skeletal muscle cells and to be able to conclude on the beneficial effect of their approach on myotube metabolism and atrophy.

C) There are thousands of protein secreted in CM from cells, which are not carried by EVs. Furthermore, use of CM for C2C12 cells is delicate as these cells are very sensitive to nutrient depletion. Therefore, to prove that the observed effect on PGC1a expression is mediated by Hsp60 overexpression in EVs and to rule out any side effect of nutrient depletion, authors should isolate EVs from CM of C2C12 cells transfected with the different plasmids, resuspend them into fresh media and treat differentiated C2C12 cells. Alternatively, authors could generate Hsp-60 enriched CM, deplete or not CM for EV (for instance by ultracentrifugation) and treat C2C12 myotubes with these two kinds of media.

D) Finally, authors should show that their approach could indeed be beneficial for treatment of cachexia. To this end, C2C12 myotubes should be treated with CM from cachexia-inducing cancer cell lines (such as C26 or LLC cells) in combination with control or Hsp-60 enriched EVs and check whether PGC1a and PGC1a target genes are upregulated. Furthermore, authors could check expression of key actors of muscle atrophy (such as E3 ubiquitin ligase or autophagy markers) and measure myotube diameter to assess the efficiency of this approach for treatment of cachexia.

Protocols for treatment of C2C12 myotubes with isolated EVs and for assessment of myotube atrophy can be found in several recent papers such as Zhang, Nat Comm, 2017 (DOI: 10.1038/s41467-017-00726-x ).

Minor comments:

  • Introduction is quite long and could be shortened. For instance, the list of drugs currently tested for treatment of cachexia is not necessary in my view.
  • Page 3 lines 110-112: statement needs to be referenced.
  • Are the levels of Hsp60 altered in plasma or muscle of cachectic mice? (example C26- or LLC-tumor bearing mice ?). Authors could include such data in the manuscript if they have access to samples or alternatively measure Hsp-60 levels in C2C12 myotubes treated with CM from cachexia-inducing cancer cell lines.
  • Materials and Methods is confusing. Some information are copy pasted between main text and supplements, or sometimes in the supplements, the text is slightly adapted. It would be better to clearly specify which kind of additional information can be found in the supplements and to remove the parts in the supplements that are identical to the main text.
  • Page 5 Line 5: which kind of starvation ? Serum starvation can affect myotube metabolism and I would not performed such a starvation. I would rather produce C2C12 CM using EV-depleted serum (commercially available or can be produced by ultra-centrifugation) and treat target cells using the same kind of media.
  • Figure 1A: please add arrows on the picture to highlight what the reader should look at
  • Figures 1-3 and the proteomic analysis, although potentially interesting as resource, does not bring a lot to the story in my view. I would rather shorten this parts of the results to include the additional experiments suggested above.

Author Response

Di Felice et al aim at producing extracellular vesicles (EV) from muscular cells carrying the chaperone protein Hsp60 for release in distant muscular cells to improve their metabolic function, and counteract diseases characterized by muscle atrophy such as cancer cachexia. To this end, they first isolated small and large EVs from C2C12 cells and performed proteomic analyses to compare their cargo. They then transfected C2C12 cells with either a control plasmid or a plasmid overexpressing the protein Hsp60. They then treated naïve C2C12 cells with conditioned media (CM) enriched in Hsp60 or not and observed an increased expression of PGC1a. Treatment of naïve C2C12 cell with human Hsp60 protein induced a similar increase in PGC1a expression. Authors therefore conclude that this approach would represent a new therapeutic tool to enhance PGC1a expression in cachectic muscle, thereby promoting mitochondria biogenesis and reducing autophagy.

The manuscript is well written and easy to follow. Although interesting, I believe the study is missing key experiments to be really convincing and to support this technique as a new potential therapeutic approach. Therefore, it would be mandatory in my view to strengthen the actual data by the following experiments:

  1. A) Key control experiments are lacking: It is mandatory to show that authors can successfully enrich Hsp60 into CM from C2C12 cells transfected with a plasmid overexpressing Hsp60 in comparison to cells transfected with a control plasmid.

Transfection efficiency is quite low (7-8%), so it is essential to show that this is sufficient to significantly enrich CM in Hsp60 protein levels.

Authors show in Figures 1-3 that they can successfully isolate small and large EVs from C2C12 CM. Therefore, I suggest that they isolate EVs from CM of cells overexpressing Hsp60 or not, and perform a western blot to confirm enrichment into secreted EVs.

That transfection of C2C12 cells increases the release of Hsp60 in conditioned medium has already been shown in detail in our previous article published in Scientific Report (Barone et al., 2016). Indeed, as detailed in Figure 8 C on page 12 of that article, we observed an increase in Hsp60 release into the medium after overexpression and a decrease in the case of siRNA against Hsp60. The transfection efficiency was similar. What we did not show is whether the Hsp60 is free or in the vesicles. However, we have conducted experiments, in the present study,  where we demonstrate that the Hsp60 is only naturally found in the Large Evs, and that the free Hsp60 does not have the same effect as the vesicles or the conditioned medium. As stated in a previous point, the small amount of culture medium that we unfortunately obtain from the transfected cells does not allow us to subdivide all types of vesicles by size or to isolate free Hsp60. However, we are working towards this issue in ongoing experiments

We are attaching a copy of the previous paper for your review.

  1. A) By treating naive C2C12 cells with CM from C2C12 cells overexpressing Hsp60 or not, authors concluded that this CM (named Physioactisome) can increase PGC1a mRNA expression and therefore may improve mitochondrial biogenesis and counteract muscle wasting for treatment of cachexia. However, this is not supported by a sufficient amount of data:

In Barone et al. (2016), we showed that exosomes rich in Hsp60 are released in serum from trained mice, after a single acute exercise session. The method of producing exosomes from serum or from medium is identical, but the volumes and number of vesicles per milliliter of medium differ significantly. In fact, in serum there are many more exosomes in small volumes, allowing us to obtain exosomes and even visualize them under the electron microscope even with a quantity of only 4 ml. However, in cell cultures we only have a few micrograms of exosomes from 200 ml of medium, and with cells that do not retain or become resistant to the plasmid, obtaining a huge amount of vesicles becomes very difficult. In previous works, we have shown, in detail, that overexpression of Hsp60 in muscle is related to training, exosomes also increase Hsp60 levels after training, and PGC-1 alpha pathway is activated under the same circumstances and in the same animals. In the present study, and also partly in our 2016 article, we attempted to mimic the same situation in vitro, both to simplify the analysis and to generate biologically active vesicles. Thus, if Physiactisome alone and without any other stimulus (differentiation in myotubes or horse serum) activates the expression of isoform 1 of PGC-1 alpha, it can be assumed that such a drug based on vesicles could have a positive effect on the muscle breakdown that characterizes pathologies such as chachessia. However, also at the suggestion of referee #1, on page 4, line 196 we will change the term "can" to "could."

  1. B) Increased PGC1a expression in response to Physioactisome (CM enriched in Hsp60 protein) has already been reported in a previous paper from the same group (Figure 8D from Barone et al, Sci Rep, 2016, doi 1038/srep19781) and is not a novel observation. Furthermore, in this paper from 2016, authors only observed an increase in PGC1a expression in target C2C12 cells after 6 hours but not after 12 hours. It seems therefore that the effect is transient and authors would actually need to show that this affect downstream targets of PGC1a to convince of the efficiency of this treatment.

The transfection method and control plasmid used in the present paper are different from the those used in our previous work (Barone et al., 2016), and here we demonstrate the release of small and large EVs from C2C12, with the expression and localization of Hsp60 both in small and large EVs.

In our previous work, we demonstrated the activation of the PGC1 pathway in trained animals. In the present work, we attempted to mimic the same situation as in vivo using C2C12 cells. Use of Physiactisome is one approach to reproduce the same situation as in vivo. Unfortunately, as mentioned previously, we do not have enough culture medium to separate different sized vesicles. We hope to address this in a future work.

  1. C) Authors use CM from undifferentiated C2C12 cells, and not from myotubes and apparently treat undifferentiated C2C12 cells to look at PGC1a expression. This is, in my view a critical point. The use of differentiated C2C12 myotubes is essential to study the metabolism of skeletal muscle cells and to be able to conclude on the beneficial effect of their approach on myotube metabolism and atrophy.

This is an excellent suggestion, but requires much experimentation and could be the subject of another publication, perhaps in conjunction with mouse muscle in vivo.

We are attempting to obtain a stable line of C2C12 cells overexpressing HSP60 so that we can not only perform WB on the treated cells but also inject the exosomes and the different vesicles into the anterior tibial muscle of Balb / c mice with and without C26 tumor (with and without cachexia). These experiments are underway and the project with animals has already been approved by the Italian Ministry of Health.

  1. D) There are thousands of protein secreted in CM from cells, which are not carried by EVs. Furthermore, use of CM for C2C12 cells is delicate as these cells are very sensitive to nutrient depletion. Therefore, to prove that the observed effect on PGC1a expression is mediated by Hsp60 overexpression in EVs and to rule out any side effect of nutrient depletion, authors should isolate EVs from CM of C2C12 cells transfected with the different plasmids, resuspend them into fresh media and treat differentiated C2C12 cells. Alternatively, authors could generate Hsp-60 enriched CM, deplete or not CM for EV (for instance by ultracentrifugation) and treat C2C12 myotubes with these two kinds of media.

These suggestions are very constructive and will certainly be incorporated in future experiments. Unfortunately, as mentioned earlier, substantial amounts of culture medium are required to obtain only a few micrograms of exosomes, and this is not possible at present. In fact, we are trying to change the plasmid or switch to adenoviruses. We are not ruling out the possibility that there is an important factor in the conditioned medium that is induced by overexpression of Hsp60, but it is not Hsp60 itself. We have some preliminary data, but they are too preliminary to be published or presented. However, in the present article, we have demonstrated the close involvement of Hsp60 using recombinant Hsp60. Thus, even at very small volumes, Hps60 has an effect. We need considerable volumes of the medium to isolate vesicles of different sizes or to perform HPLC of the medium to understand which molecule might have an effect on PGC_1 alpha signaling by mimicking movement.

  1. E) Finally, authors should show that their approach could indeed be beneficial for treatment of cachexia. To this end, C2C12 myotubes should be treated with CM from cachexia-inducing cancer cell lines (such as C26 or LLC cells) in combination with control or Hsp-60 enriched EVs and check whether PGC1a and PGC1a target genes are upregulated. Furthermore, authors could check expression of key actors of muscle atrophy (such as E3 ubiquitin ligase or autophagy markers) and measure myotube diameter to assess the efficiency of this approach for treatment of cachexia.

This is also an excellent suggestion, which we will certainly consider for future experiments.

  1. F) Protocols for treatment of C2C12 myotubes with isolated EVs and for assessment of myotube atrophy can be found in several recent papers such as Zhang, Nat Comm, 2017 (DOI: 10.1038/s41467-017-00726-x ).

Minor comments:

Introduction is quite long and could be shortened. For instance, the list of drugs currently tested for treatment of cachexia is not necessary in my view.

Response: List deleted, and paragraph edited as per comment. See page 2 line 94 – 99 of the marked version of the paper.

Page 3 lines 110-112: statement needs to be referenced.

We have added a reference of a very recent review on the effects of exercise on colon cancer progression at page 3, line 113.

Are the levels of Hsp60 altered in plasma or muscle of cachectic mice? (example C26- or LLC-tumor bearing mice ?). Authors could include such data in the manuscript if they have access to samples or alternatively measure Hsp-60 levels in C2C12 myotubes treated with CM from cachexia-inducing cancer cell lines.

We thank the reviewer for these suggestions. Unfortunately, we do not have serum from cachectic animals because we used it for another study, but we do have the immunohistochemical findings showing the expression of Hsp60 protein in the Tibialis anterior muscle of cachectic mice and their controls, as well as a positive control for Hsp60 expression in a section of human colon carcinoma (the antibody recognizes both human and mouse proteins). We added the file only to comply with the reviewer's request, but we do not wish to publish it as supplementary material, since the cachectic animal model and a possible anti-cancer effect are not the subject of this study. In fact, our product has positive effects on PGC-1 alpha pathway  and muscle regeneration and no anti-tumor effect.

We have included the immunohistochemistry showing the expression of Hsp60 in the Tibialis Anterioris to elucidate the results of our past experiments. We shall continue to work to publish more results.

We have no data on the effect of Physiactisome on tumor cells, but since C26 tumor cells overexpress Hsp60 as a tumor marker, we do not expect Hsp60 expression levels to change.

Materials and Methods is confusing. Some information are copy pasted between main text and supplements, or sometimes in the supplements, the text is slightly adapted. It would be better to clearly specify which kind of additional information can be found in the supplements and to remove the parts in the supplements that are identical to the main text.

As per the suggestion, we have specified what is described in the Supplementary Information and deleted the parts that were redundant or repeated between the main text and the Supplementary Information. (See Materials and Methods section).

Page 5 Line 5: which kind of starvation ? Serum starvation can affect myotube metabolism and I would not performed such a starvation. I would rather produce C2C12 CM using EV-depleted serum (commercially available or can be produced by ultra-centrifugation) and treat target cells using the same kind of media.

At page 4 line, 201 there was a proofreading error on our part, which has been rectified. We used ultracentrifuged FBS (made by ourselves) for all the experiments. It was missing the time from transfection. Therefore, we changed the phrase in “After 72 h from transfection….,” added the word “ultracentrifuged” in the Supporting Information regarding cell cultures.

Figure 1A: please add arrows on the picture to highlight what the reader should look at

Figures 1-3 and the proteomic analysis, although potentially interesting as resource, does not bring a lot to the story in my view. I would rather shorten this parts of the results to include the additional experiments suggested above.

We have added arrows to Figure 1. Figure 2 has been replaced with a similar but clearer experiment, and signs have been added to guide the readers. Figure 3 already has arrows.

Regarding proteomic analysis, this was conducted to demonstrate once again the presence of Hsp60 in vesicles, since C2C12 express limted Hsp60 under basic conditions and it is difficult to detect it by western blotting. Moreover, to our knowledge, this is the first published proteomic study describing the protein content of large vesicles belonging to C2C12 cell line.

With respect to the proposed experiments, we have added relevant details in the responses pasted above.

Reviewer 3 Report

In this preclinical study, the authors aim to obtain a conditioned medium released by Hsp60–overexpressing C2C12 cells enriched with small and large extracellular vesicles (Physiactisome), capable of activating PGC-1α, particularly the isoform 1 in naïve C2C12 cell line and mouse immortalized myoblasts. Authors show that Physiactisome activates PGC-1α isoform 1 expression and recapitulates the conditions prevalent during the muscle response in trained mice and inducing the same level of PGC-1α expression as that observed in vivo. Finally, authors suggest Physiactisome use in the treatment of cachexia and all conditions characterized by muscle wasting and damage.

In my opinion, the results are clearly displayed both in the text and in the figures, but I have some doubts.

Line 87: “Currently, none of the commercially available drugs effectively reverse cachexia or counteract muscle wasting and loss of lean mass…..” Authors should specify that there are already studies in the literature in which it is shown that molecules are able to reduce muscle alterations characterizing the cachectic muscle (doi: 10.1038/s41598-017-13504-y; 10.3892/mmr.2022.12610 ). Furthermore, in the first article, it is also shown that the treatment causes an increase in PGC-1α expression.

Line 524: “In conclusion, our study suggests that Physiactisome treatment of C2C12 cells recapitulates the conditions prevalent during the muscle response in trained mice and induces the same level of PGC-1α expression as that observed in vivo”. Considering that this study is an in vitro study, I think this sentence represents strong speculation if the same results are not confirmed in vivo.

For real-time experiments, why did you choose GAPDH as housekeeping? In my experience, it is not the most stable housekeeping. Did you analyze any others before choosing gapdh?

Author Response

We thank the reviewer for revising and improving our work through his/her comments. We appreciated his/her help in improving our work.

In this preclinical study, the authors aim to obtain a conditioned medium released by Hsp60–overexpressing C2C12 cells enriched with small and large extracellular vesicles (Physiactisome), capable of activating PGC-1α, particularly the isoform 1 in naïve C2C12 cell line and mouse immortalized myoblasts. Authors show that Physiactisome activates PGC-1α isoform 1 expression and recapitulates the conditions prevalent during the muscle response in trained mice and inducing the same level of PGC-1α expression as that observed in vivo. Finally, authors suggest Physiactisome use in the treatment of cachexia and all conditions characterized by muscle wasting and damage.

In my opinion, the results are clearly displayed both in the text and in the figures, but I have some doubts.

Line 87: “Currently, none of the commercially available drugs effectively reverse cachexia or counteract muscle wasting and loss of lean mass…..” Authors should specify that there are already studies in the literature in which it is shown that molecules are able to reduce muscle alterations characterizing the cachectic muscle (doi: 10.1038/s41598-017-13504-y; 10.3892/mmr.2022.12610 ). Furthermore, in the first article, it is also shown that the treatment causes an increase in PGC-1α expression.

We thank the reviewer for finding these interesting papers that further validate our results and are really interesting to continue our project and work.

We have accepted the suggestion to comment and add these two recent references, and we changed the introduction accordingly. Newly added text has been coloured blue.

Page 2 lines 87-92.

Line 524: “In conclusion, our study suggests that Physiactisome treatment of C2C12 cells recapitulates the conditions prevalent during the muscle response in trained mice and induces the same level of PGC-1α expression as that observed in vivo”. Considering that this study is an in vitro study, I think this sentence represents strong speculation if the same results are not confirmed in vivo.

To reduce speculation in this sentence, we decided to delete the following words: “and induces the same level of PGC-1α expression as that observed in vivo”. Page 15, line 539 – 540.

For real-time experiments, why did you choose GAPDH as housekeeping? In my experience, it is not the most stable housekeeping. Did you analyze any others before choosing gapdh?

Yes, we used also GUSB and Hprt as housekeeping, but we noticed that only GADPH was the more stable in our samples and conditions.

Reviewer 4 Report

Authors performed an interesting and well documented project on a original method to produce an exercise-mimicking drug, with possible important implication for translation medicine and drug discovery. The work is valuable, but some important points should be addressed in the opinion of this reviewer

General comments

One major point is that authors missed to explain why their method should be preferred over other biotech procedures to produce EVs rich in selected molecules

Another major point is that the quality of purification of EVs may be questionable in a high-throughput drug fabrication framework and that it cannot be completely excluded that contaminants may have biased the proteomic results. Authors should at least discuss this limit and suggest possible implementation technique

Overall, authors always refer to nanovesicles, without a definition of that term; although they correctly use the term nanovesicles, considering the discussion in categorizing EVs (see e.g. https://doi.org/10.3390/ijms22126417) they should at least include a definition (see e.g. https://doi.org/10.1016/j.biopha.2021.112416).

Specific comments

Lines 66-68: this part needs a revision, paying attention to radical vs reactive species definitions; authors state that radicals may interfere to produce other reactive products, e.g., hydroxyl; however, it may derive from non-radicals such as hydrogen peroxide; therefore, authors should not use interchangeably reactive species and radicals

Line 114: the effect of resistance training on IL6 should be cited with caution, as this cytokine may have a dual (i.e., beneficial vs detrimental) role as a molecular determinant of exercise effect; in fact, authors furtherly describe an acute positive effect in line 126; ref. 11 might not be an adequate reference, at least if cited alone; see e.g. https://doi.org/10.1152/ajpregu.00147.2019

Line 142-143: authors say that this finding is in agreement with the above-mentioned studies; however, this seems to disagree with ref 19 (line 135)

Line 245: why did not the authors meaured also zeta potential with the Zetasizer? It may provide additional and meaningful info about the features of released vesicles, considering one of the authors' aim is to suggest this novel drug fabrication method

Line 318: "sub-membranous"?

LInes 339-350: why did not the author assess the presence/absence of expected contaminants, as negative markers, for improving quality control?

Line 438: please write the exact p value

Line 537: concluding the manuscript with a reference sounds weird

Line 630: reference 25 is missing

Author Response

As for Reviewer #3, we thank Reviewer #4 for the revision and for improving our paper with his/her comments.

We appreciated his/her work and have done everything possible to accomplish his/her suggestions.

Authors performed an interesting and well documented project on a original method to produce an exercise-mimicking drug, with possible important implication for translation medicine and drug discovery. The work is valuable, but some important points should be addressed in the opinion of this reviewer

General comments

One major point is that authors missed to explain why their method should be preferred over other biotech procedures to produce EVs rich in selected molecules

We have added the following paragraph to page 15 lines 530 – 536.

“This method may be preferable to direct manipulation of extracellular vesicles because it preserves the membrane lipid composition and protein content of vesicles released from muscle cells. Although attempts can be made to obtain the manipulated vesicles from cells other than muscle cells (e.g., cancer cells), we believe that for proper uptake of the manipulated vesicles from the target cell, it is important to choose the same type of cell as the target cell. This can be demonstrated by extracting green vesicles from GFP-expressing cells and treating naive cells.”

Another major point is that the quality of purification of EVs may be questionable in a high-throughput drug fabrication framework and that it cannot be completely excluded that contaminants may have biased the proteomic results. Authors should at least discuss this limit and suggest possible implementation technique

On page 9 in the paragraph from line 339 to 350, we described how the expression of Hsp70, RAB5, Alix, and calnexin indicates the good quality of the separation method. These experiments are necessary before we proceed with the time-consuming and expensive proteomic analysis. Of course, there may be contaminants. To bring this method to market, we need to perform a size purification step before ultracentrifugation, we are trying to achieve good results with chromatographic purification techniques.

According to the reviewer suggestion we discussed this point.

We added in the discussion section the following sentences at page 15 lines 507 – 515.

One of its limitations is that the molecular method for the overexpression of Hsp60 and EVs purification must be further improved to obtain stable transfection and Physiacti-some-producing C2C12 cell line to scale-up production. Methods to isolate an EV sub-population are subjects of constant debate and, although ultracentrifugation is the most commonly used, others techiques, such as size exclusion chromatography (SEC) using different matrices (e.g., Sepharose 2B, Sephacryl S-400), filtration or precipitation have been utilized (PMID 26690353). More recently, a combination of different methods (eg. ultracentrifugation and SEC) are employed for improving the purity of EVs (PMID 32349218).

Overall, authors always refer to nanovesicles, without a definition of that term; although they correctly use the term nanovesicles, considering the discussion in categorizing EVs (see e.g. https://doi.org/10.3390/ijms22126417) they should at least include a definition (see e.g. https://doi.org/10.1016/j.biopha.2021.112416).

Also in this case we appreciated the referee's comments and tried to answer the question by inserting the following paragraph on page 4 lines 154 – 162.

“Our EVs are naturally secreted, biolipid-based, cell-derived nanovesicles, which should be more biocompatible and host-friendly compared to artificial nanoparticle-based nano-carriers (PMID 34781147). Although the literature is full of articles discussing the nomenclature of extracellular vesicles, such as the recent revision of their historical nomenclature (PMID 34203956), we chose to define our vesicles as nanovesicles because we, as morphologists, recognize them as nanosized vesicles. Based on the population of vesicles found with the Zetasizer, we then distinguished nanovesicles into small extra-cellular vesicles if they are between 80 and 200 nm in size and large extracellular vesicles if they are larger than 200 nm.”

Specific comments

Lines 66-68: this part needs a revision, paying attention to radical vs reactive species definitions; authors state that radicals may interfere to produce other reactive products, e.g., hydroxyl; however, it may derive from non-radicals such as hydrogen peroxide; therefore, authors should not use interchangeably reactive species and radicals

The reviewer is right, we misused the term radicals as if it were the same as reactive species, so we removed the word "radicals" on page 2 line 67.

Line 114: the effect of resistance training on IL6 should be cited with caution, as this cytokine may have a dual (i.e., beneficial vs detrimental) role as a molecular determinant of exercise effect; in fact, authors furtherly describe an acute positive effect in line 126; ref. 11 might not be an adequate reference, at least if cited alone; see e.g. https://doi.org/10.1152/ajpregu.00147.2019

In line 105 we have cited one of our articles in which we discussed the role of IL6 in the terms indicated by the referee. As suggested we have added to the old citation 11, now 13, the citation indicated (n. 14).

Line 142-143: authors say that this finding is in agreement with the above-mentioned studies; however, this seems to disagree with ref 19 (line 135)

There is no discrepancy with citation n.19 because in the paragraph it says that not all muscles overexpress the Hsp60, but the soleus in our case does. The two references are about different muscles.

Line 245: why did not the authors meaured also zeta potential with the Zetasizer? It may provide additional and meaningful info about the features of released vesicles, considering one of the authors' aim is to suggest this novel drug fabrication method

We regret not having understood which line the reviewer is referring to, let's suppose that he is referring to the hypothesis of doing the Zetasizer also on Physiactisome. Unfortunately, as we have often stated, the quantity of vesicles and conditioned medium isolated from engineered C2C12 cells has not yet allowed us to make this determination as well.

Line 318: "sub-membranous"?

It has been referred to as sub-membranous because if Alix is an intracellular protein, the radial arrangement of the antibodies bound to the gold nanoparticles can only indicate the localization below the membrane.

LInes 339-350: why did not the author assess the presence/absence of expected contaminants, as negative markers, for improving quality control?

According to the reviewer suggestion we added the following sentence to page 11 lines 388-390: “Moreover, APOA1/2 and Albumin, generally co-isolated with EVs structures as contaminants were absent, suggesting the good quality of our preparations”.

Line 438: please write the exact p value

Done

Line 537: concluding the manuscript with a reference sounds weird

We have deleted the last reference n. 36 which was not so important.

Line 630: reference 25 is missing

Done.

Round 2

Reviewer 1 Report

The manuscript is modestly improved.  The authors did not correct the sentence on page 4, line 161, which implies they worked on two cell lines.  This sentence must be corrected.

Author Response

None

Reviewer 2 Report

I thank the authors for their answers to my questions, however I am still not convinced about the meaning of the study.

1) Concerning the key control experiment showing that Hsp60 is enriched in conditioned media from C2C12 cells transfected with an Hsp60-oeverexpression plasmid, authors said that this has already been published in Barone et al., 2016, Figure 8C. However, they later say that the transfection method and control plasmid used was different, so this is not valid. Authors actually have to show the overexpression. They explain that they do not have enough EV from media to perform a western blot showing Hsp60-overexpression (although they actually show a WB in Figure 3 of the actual paper) but at least I would need to see the same kind of measurement as in Barone et al., 2016.

2) The fact that authors use undifferentiated C2C12 cells is really an issue for me as they do not represent muscle fibers. I therefore do not understand what the observation means for muscle function.

3) As mentioned in the previous report, the increase in PGC1a expression in C2C12 cells treated with conditioned media enriched in Hsp60 protein is not new, it has already been published in Barone et al., 2016, Figure 8C & D. In this previous paper, the treatment is even more convincing as the control experiment showing an enrichment of Hsp60 in conditioned media is presented in Figure 8C. What is new is the proteomic analysis of EV released by undifferentiated C2C12 cells but these information are not major in my view as these are from undifferentiated C2C12 cells and not even from myotubes, which are representative of differentiated muscle cells.

Because of all these major concerns, I do not think that this paper is suitable for publication in a journal with an impact factor >6. I think the study is interesting and has some potential in the field in cachexia, but that would require to perform the experiments suggested in the previous report.

Author Response

None